# Biomarkers for Kidney-Transplant Rejection: A Short Review Study

**DOI:** 10.3390/biomedicines11092437

**Published:** 2023-08-31

**Authors:** Israa Sharaby, Ahmed Alksas, Mohamed Abou El-Ghar, Mona Eldeeb, Mohammed Ghazal, Dibson Gondim, Ayman El-Baz

**Affiliations:** 1Bioengineering Department, University of Louisville, Louisville, KY 40292, USAammost01@louisville.edu (A.A.); 2Radiology Department, Urology and Nephrology Center, Mansoura University, Mansoura 35516, Egypt; maboelghar@mans.edu.eg (M.A.E.-G.); mona.ahmed@mans.edu.eg (M.E.); 3Electrical, Computer, and Biomedical Engineering Department, Abu Dhabi University, Abu Dhabi 59911, United Arab Emirates; mohammed.ghazal@adu.ac.ae; 4Department of Pathology and Laboratory Medicine, University of Louisville, Louisville, KY 40202, USA; dibson.gondim@louisville.edu

**Keywords:** acute-mediated rejection (AMR), biomarkers, histopathological markers, kidney-transplant rejection, renal rejection

## Abstract

Kidney transplantation is the preferred treatment for end-stage renal failure, but the limited availability of donors and the risk of immune rejection pose significant challenges. Early detection of acute renal rejection is a critical step to increasing the lifespan of the transplanted kidney. Investigating the clinical, genetic, and histopathological markers correlated to acute renal rejection, as well as finding noninvasive markers for early detection, is urgently needed. It is also crucial to identify which markers are associated with different types of acute renal rejection to manage treatment effectively. This short review summarizes recent studies that investigated various markers, including genomics, histopathology, and clinical markers, to differentiate between different types of acute kidney rejection. Our review identifies the markers that can aid in the early detection of acute renal rejection, potentially leading to better treatment and prognosis for renal-transplant patients.

## 1. Introduction

Kidney transplantation is a superior treatment to dialysis for individuals with chronic kidney disease or end-stage renal failure, boasting up to 97% survival rates for transplanted kidneys within one year [1]. Nevertheless, a significant hurdle is the limited availability of donors, mainly due to the risk of immune rejection. When genetically dissimilar donor tissue is transplanted, the recipient’s immune system may perceive it as foreign, leading to potential graft rejection [2].

Lymphocytes, essential white blood cells, are vital components of the adaptive immune response, with B-cells producing pathogen-specific antibodies and T-cells capable of killing infected cells or seeking support from other cells [3]. The interaction between the innate and adaptive immune systems defends the body against foreign pathogens and abnormal cells [2,3]. Recent advancements in technology have enabled personalized immunosuppressive therapies based on recipient-specific biomarkers related to immune response activation [3]. *Toll-like receptors (TLRs)* are extensively studied pattern-recognition receptors (PRRs) that play a critical role in initiating innate responses and guiding adaptive immunity [4]. They are expressed in various hemopoietic cells, including DCs, B-cells, mast cells, T-cells, and endothelial cells. Their stimulation leads to the activation of transcription factors NF-KB and AP-1, which subsequently induce the transcription of inflammation-related genes. This results in the production of proinflammatory cytokines, chemokines, antimicrobial peptides, adhesion molecules, enhanced antigen presentation, and increased expression of costimulatory molecules in APCs. The intricate interplay among these components significantly influences the immune response against transplanted organs and tissues, ultimately impacting the rejection outcome [4].

Acute renal rejection, classified based on histopathological and immunological characteristics, has distinct forms [2]. Hyperacute rejection is a severe type that occurs suddenly within minutes of transplant [5], but it is rare due to successful prevention through tissue cross-matching. Acute rejection can happen at any time after transplantation and includes antibody-mediated rejection (ABMR) and T-cell-mediated rejection (TCMR). ABMR involves immunological damage caused by antibodies and donor-specific alloantibodies in circulation, while TCMR results in lymphocytic infiltration affecting the interstitium, tubules, and sometimes the artery intima. Tubular injury markers differentiate between ABMR and TCMR based on specific histological features. For instance, TCMR is characterized by mononuclear tubulitis and interstitial inflammation, while ABMR exhibits microvascular inflammation, arteritis, acute tubular injury, or thrombotic microangiopathy. Additionally, the presence of complement component C4d in peritubular capillaries indicates the antibody–endothelium interaction in ABMR, although its reproducibility is poor and staining results may vary [2].

The prohibitive expense of postsurgical maintenance, amounting to thousands of dollars per month for antirejection medications, presents a significant challenge for some patients [6]. Untreated rejection can lead to major health problems [7]. To address this issue and enhance transplant outcomes, early detection of renal rejection through innovative biomarkers is essential [8]. These markers include histological, clinical, and genetic indicators. While biopsy remains the gold standard, from these markers, clinical biomarkers such as *KIM-1* and *CXCL-10* show potential for early diagnosis and prognosis prediction of renal rejection [9].

The aim of this short review is to provide an overview of the current state of biomarkers for acute renal rejections and their correlation with the type of acute renal rejections (refer to Figure 1). In addition, we will highlight which markers are noninvasive and may lead to the development of new noninvasive technology for the early detection of acute renal rejection and its associated type.

This review presents the state of the art regarding markers used to detect renal rejection. It is structured as follows: Section 2 provides an overview of studies utilizing histopathological markers, Section 3 discusses studies employing clinical markers, and Section 4 examines studies using genetic markers. The review highlights the strengths and limitations of the reviewed studies in Section 5 and presents overall conclusions and future research directions in Section 6. Data regarding the three types of markers and their usage for determining renal rejection were collected through a search of articles over the last decade.

## 2. Extraction of RNA and Clinical Markers from Histological Samples

Histopathological markers refer to changes in tissue structure that are observed through a microscope, and they aid in identifying and monitoring renal-transplant rejection. These markers offer valuable insights into the cellular and molecular changes that occur during rejection and can be challenging to detect with clinical markers alone [10]. The presence of infiltrating lymphocytes, which can be detected through a biopsy sample taken from the transplanted kidney, is one of the most commonly used histopathological markers of renal rejection. It indicates an active immune reaction against the grafted tissue. The severity of rejection can be graded based on the number and distribution of infiltrating lymphocytes, with higher numbers indicating more severe rejection [10,11]. Other histopathological markers of renal rejection include changes in the capillary basement membrane, alterations in the glomerular filtration barrier, and modifications in the tubulointerstitium. These changes may indicate acute tissue injury and early lesions of endothelial injury and can be visualized using light microscopy or electron microscopy [12].

Table 1 utilizes histopathological measures and immune markers to identify acute renal rejection and confirm the effectiveness of the selected markers. A study conducted by Arai et al. [13] used light and electron microscopy to detect histological evidence of acute tissue and endothelial injury in 269 kidney-transplant recipients. The results showed no worsening of clinical or microscopic features in patients with short-term follow-up. Similarly, in a cross-sectional study, Garcia-Covarrubias et al. [14] examined the levels of interleukin in urine samples and biopsies from 37 kidney-transplant recipients, including both rejected and nonrejected patients, as well as a group of healthy individuals. The primary objective of the study was to determine whether there were any variations in interleukin expression levels across the groups. However, when comparing the transplant patients who had been rejected and those who had not with the healthy group, the interleukin analysis did not show any appreciable differences. The findings of both studies raise questions about the reliability and significance of these markers in predicting rejection outcomes.

In their meta-analysis, Eiamsitrakoon et al. [15] aimed to investigate the effect of the *IFNG rs2430561* gene on the likelihood of acute rejection in transplant patients. The study found a significant increase in the risk of rejection in only the black subgroup. However, the results were deemed weak due to a low sample size and a wide confidence interval. Rohan et al. [16] conducted a study to assess the efficacy of allospecific CD154+ TcM in 22 individuals who received kidney transplants. Out of the 22 patients, 6 experienced TCMR and 7 showed ABMR. Among the 11 patients who tested positive, 6 experienced acute cellular rejection (ACR) or antibody-mediated rejection (ABMR), whereas 10 out of the 11 patients who tested negative did not encounter rejection.

Shimizu et al. [17] analyzed 22 renal-allograft-biopsy specimens from 20 transplant recipients to investigate the clinical and pathological significance of borderline changes (BCs) after transplantation. BC was diagnosed around 500 days post-transplantation and was characterized by tubulitis, interstitial inflammation, peritubular capillaritis, and varying degrees of interstitial fibrosis and tubular atrophy. Glomerulitis and intimal arteritis were not observed. While no graft loss occurred, 45% of patients experienced deterioration in renal-allograft function. The findings highlight the importance of monitoring and treating BCs appropriately. A cross-sectional study by Zhuang et al. [18] aimed to evaluate B-cell subsets in immunologically stable renal-transplant patients and those with end-stage renal disease (ESRD). Flow cytometry was used to analyze B-cell subsets in 73 stable transplant recipients, 35 ESRD patients, and 36 healthy volunteers. The findings revealed lower percentages of total B-cells and regulatory B-cells in transplant recipients compared to healthy controls. Transitional and marginal-zone B-cells were also reduced in transplant recipients compared to ESRD patients and healthy volunteers. ESRD patients exhibited higher percentages of plasma cells. Overall, B-cell subsets differed significantly between immunologically stable renal-transplant recipients and healthy controls, with minimal differences observed between ESRD patients and transplant recipients.

Mueller et al. [19] conducted an RNA sequencing study on 34 kidney allograft biopsies to investigate differences between normal biopsies and those exhibiting T-cell-mediated rejection. The study revealed that biopsies with T-cell-mediated rejection had higher mRNA levels for pattern-recognition receptors and calcineurin, indicating reduced immunosuppression compared to healthy biopsies. Nowanska et al. [20] used immunohistochemistry techniques to examine the presence of endothelin receptors (ETARs) in 162 kidney-transplant-biopsy samples. The study found that patients who tested positive for ETARs had a higher rate of graft loss one year after the biopsy, but the findings were not statistically significant.

De Leur et al. [21] explored the control of B-cell-mediated immunity in acute T-cell-mediated rejection by examining 15 biopsy samples from kidney-transplant recipients. The study found that 40% of the biopsies contained dense cellular aggregates. Biopsies displaying acute T-cell-mediated rejection (aTCMRI) had these clusters in 80% of cases, while the aTCMRII and a/aABMR groups only exhibited them in 20% of cases. Wu et al. [22] performed a study that employed single-cell RNA sequencing to identify and classify different cell types and conditions present in a human kidney biopsy. The study examined 8746 individual cell transcriptomes from a healthy adult kidney and a solitary kidney transplant. Upon histological inspection, the biopsy of the transplant exhibited acute T-cell-mediated rejection with plasma cells evaluated as Banff 1B, along with acute C4d-negative antibody-mediated rejection (ABMR).

McRae et al. [23] investigated the expression of CD4+ CD25+/CD39+ in the peripheral blood of 17 end-stage renal-failure patients. The study found that patients with acute T-cell-mediated rejection had lower levels of CD4+ CD25+/CD39+ mTreg and CD4+ CD25+ CD39+ mTeff cells compared to nonrejected patients as determined by flow cytometry analysis. The objective of Jiang et al. [24] was to identify CD20-positive B-cell effectiveness in biopsy specimens from 216 patients. The findings indicated that, out of the total participants, 83 patients were categorized as belonging to the CD20-negative group, while 133 patients were classified as part of the CD20-positive group.

Sentis et al. [25] aimed to predict graft failure that is censored for death by examining the count of CD68+ macrophage glomerular cells during an incident of acute rejection in 57 patients who underwent renal transplantation. The study found that 42% of patients who experienced acute rejection lost their graft after a median of 1236 days, and concluded that CD68+ macrophage cell count can predict graft failure that is censored for death. Matingon et al. [26] conducted research on 43 kidney-transplant-biopsy samples to assess the alloimmune responses of Th-17. Using molecular analysis, the study identified patients who expressed mRNA IL17 but had low-mRNA Foxp3 expression. These patients had a much higher rate of treatment failure (87.5%) compared to those who did not express mRNA IL17 or had high-mRNA Foxp3 expression (26.7%, *p* = 0.017).

Visona et al. [27] investigated the role of lymphomononuclear inflammation (nephritis) in kidney transplants and were involved in the analysis of 113 kidney-transplant biopsies.

According to the study, 66 cases (58.4%) of the analyzed kidney-transplant biopsies had nephritis in both the medulla and cortex, while 47 cases (41.6%) showed nephritis exclusively in the medulla. The findings indicate that the existence of medullary nephritis in kidney-transplant biopsies could indicate the occurrence of acute cellular rejection. Lee et al. [28] examined the effect of anti-AT1R antibodies on renal-allograft rejection by analyzing 53 patients who were diagnosed with rejection through biopsy. This included 22 patients with antibody-mediated rejection, 29 with T-cell-mediated rejection, and 2 with mixed rejection. According to the study, anti-AT1Rs were detected in 9.4% of the patients who experienced rejection, and this was strongly linked with HLA class-I DSAs.

Crespo et al. [29] investigated the connection between peripheral blood NK-cell subsets, clinical characteristics, and circulating anti-HLA antibodies (DSAs and non-DSAs) in a group of 393 recipients of renal allografts. After performing multivariate analysis, the study found that patients without detectable anti-HLA antibodies had lower levels of NK cells but had a more significant rise in CD56 bright and NKG2A+ subsets, particularly in cases with DSAs. In a separate study, Bhat et al. [30] aimed to examine the function of p-S6RP plasma cells in instances of acute renal-allograft rejection. The research involved analyzing a group of 28 patients to determine their response to antirejection treatment, measured by the serum creatinine ratio. The findings revealed that patients with high-p-S6RP staining had a significantly higher creatinine ratio compared to those with low-p-S6RP staining, indicating a less favorable response to treatment. These outcomes suggest that p-S6RP plasma cells play a role in acute rejection.

De Vos et al. [31] examined the monitoring of de novo DSAs (dDSAs) in a substantial group of renal-transplant recipients from multiple ethnicities. Through a nested case-control study, the research found that, among the 503 recipients, 24% developed dDSAs, with 73% of those individuals having dDSAs directed against the DQ antigen. Recipients with dDSA had a greater probability of encountering various types of acute rejection, such as antibody-mediated acute rejection (16%), acute rejection caused by noncompliance (8%), and recurrent acute rejection (6%), compared to those without dDSAs. Additionally, they had a higher chance of experiencing acute rejection overall (35%). In another study by Zhao et al. [32], the mechanism by which CD4+ T-cells secrete *sFGL2* in the onset of acute rejection was explored in a sample of 40 cases. The study concluded that renal-allograft recipients with acute rejection confirmed through biopsy had significantly higher levels of *sFGL2*, TNF-α, IFN-γ, and CD4+ T-cells in their peripheral blood.

Ge et al. [33] conducted a meta-analysis on 525 cases of acute renal-allograft rejection to investigate the correlation between the *IFNG +874 T* > A polymorphism and susceptibility to AR. The research found a significant association between possessing the T allele and increased susceptibility to AR. The correlation was particularly notable in Caucasians and individuals who received a kidney transplant from a cadaveric donor. Another study by Loupy et al. [34] explored the relationship between the ability of anti-HLA antibodies to bind to complement and kidney-transplant failure. The study analyzed 1016 patients for circulating anti-HLA antibodies and revealed that patients with anti-HLA antibodies capable of binding to the complement had a much lower rate of graft survival (54%) compared to those without these antibodies (93–94%).

Li et al. [35] investigated the use of immunophenotyping to distinguish between BK virus nephropathy and acute rejection in 65 kidney-transplant recipients. The study found that quantities of CD3, CD4, CD8, and CD20 cells, as well as the count of CD20 cells detected in renal biopsies, could aid in distinguishing between the two conditions. In another study, Xu et al. [36] examined the immune phenotype of T-lymphocyte infiltrations in renal biopsies taken from 125 transplant recipients with stable renal function. The research revealed that individuals with a regulatory T-lymphocyte phenotype had a lower frequency of acute rejection (83.2%) compared to those with a cytotoxic T-lymphocyte phenotype, where all instances resulted in clinically diagnosed or biopsy-proven acute rejection.

Chang et al. [37] conducted a study on 56 sequential biopsies of renal transplants to predict the outcomes of grafts after acute rejection. They analyzed plasma cell densities and glomerular filtration rates and found that plasma cell density was a crucial predictor of graft failure. The research also showed a trend towards statistical significance to B-cell density.

In summary, the studies mentioned in the text provide diverse results regarding the detection of kidney-transplant rejection. While some histopathological and clinical markers show promise for predicting rejection, there are conflicting findings among the studies, possibly due to variations in sample sizes, study designs, and patient characteristics. The lack of information on the replication and validation of findings raises uncertainty about the overall reliability of some results. Larger sample sizes in some studies enhance the credibility of the conclusions, but smaller sample sizes in others may limit the statistical power. The discussed markers have potential applications in clinical practice, but further validation and research in larger and diverse patient populations are needed to ensure their reliability and widespread implementation. Healthcare providers must keep abreast of evolving research to make evidence-based decisions in diagnosing and monitoring renal-transplant rejection.

The mentioned studies investigated different types of rejection in kidney-transplant recipients. A common finding among these studies was the importance of T-cell mediated rejection (TCMR) and antibody-mediated rejection (ABMR) in acute rejection [16,17,19,20,21,26]. Several studies identified the biomarker CD4+ CD25+/CD39+ expression to predict acute cellular rejection (ACR) as well [23,24,27]. The studies also highlighted the importance of detecting and treating acute rejection, as it was found to increase the risk of graft loss [15,31]. Additionally, they identified various risk factors for acute rejection, including endothelial injury, low levels of CD4+ CD25+/CD39+ mTreg and CD4+ CD25+ CD39+ mTeff cells, and the alloimmune responses of Th-17 [22,23,32]. These histological samples emphasize their critical importance in identifying different types of renal rejection, which provide valuable insights into the mechanisms underlying kidney-transplant rejection and may help improve treatment outcomes for kidney-transplant patients.

## 3. Clinical Markers

Clinical markers play a critical role in identifying and monitoring the progression of renal rejection, allowing for timely intervention to prevent further damage to the transplanted kidney. Common clinical markers used to detect renal rejection include serum creatinine levels, blood urea nitrogen (BUN) levels, urine output, and proteinuria [37]. An increase in serum creatinine levels may suggest a decrease in kidney function, potentially indicating the onset of rejection. Elevated BUN levels can also indicate a reduction in kidney function. Decreased urine output may be a sign of decreased kidney function or possible obstruction of the urinary tract, both of which can be linked to rejection [38]. Proteinuria, which refers to an excessive amount of protein in the urine, may indicate damage to the glomeruli, the small filtration units within the kidney. Despite having a critical role in identifying rejection, these markers have some limitations detecting renal rejection. They are influenced by nonrenal factors, making early detection of rejection challenging [39]. To address these limitations, researchers have explored other biomarkers, such as interleukin-18 (IL-18) and neutrophil-gelatinase-associated lipocalin (NGAL), released by injured kidneys, which show promise in providing early and sensitive detection of kidney-transplant rejection [39]. It is essential to note that these clinical markers are usually used in combination with other tests, such as biopsy or imaging, to confirm a rejection type and diagnosis, in addition to allowing real-time monitoring of the transplanted kidney. With early detection and prompt treatment, the prognosis for most cases of renal rejection is favorable, and the transplanted kidney can often be salvaged [40]. Additional markers include: urinary *β*2-microglobulin, N-acetyl-*β*-glucosaminidase (NAG), and L-FABP, which are markers of rejection in urine tests used in real clinical practice, with *β*2-microglobulin assessing proximal tubule injury [41], NAG serving as a sensitive marker of tubular injury [37], and L-FABP being a valuable biomarker for diagnosing acute kidney injury and predicting long-term graft outcomes in kidney-transplant patients [42]. Additionally, the researchers investigated other clinical markers that may be correlated with acute renal rejection, and their studies and findings will be highlighted below.

Heidari et al. [43] conducted a study to examine a novel biomarker to diagnose antibody-mediated rejection (ABMR) in 36 patients who had undergone kidney transplants. The study utilized an RF algorithm to identify a panel of three proteins, namely EGF, *COL6A*, and *NID-1*, that demonstrated potential as a useful method for identifying ABMR early on, with high accuracy and precision.

Zhang et al. [44] conducted a study on 282 patients from a public repository of the high-throughput Gene Expression Omnibus database at the National Center of Biotechnology Information that identified four specific long noncoding RNAs (*lncRNAs*) that could potentially serve as biomarkers to diagnose acute rejection (AR) and predict the likelihood of kidney-transplant failure. These *lncRNAs* are *ATP1A1-AS1*, *LINC00645*, *EMX2OS*, and *CTD-3080P12.3*. By using univariate and multivariate Cox regression analyses, the researchers developed a *4-lncRNA* risk score model based on 17 prognostic *DElncRNAs,* indicating the potential of these *lncRNAs* as diagnostic and prognostic biomarkers. Similarly, Nolan et al. [45] examined the efficacy of the urinary Q-Score in detecting the acute rejection of renal allografts. They collected 223 urine samples from patients of all ages who had undergone renal transplants and measured 6 QSant biomarkers. The statistical models developed achieved a 99.8% ROC and 98.2% accuracy, suggesting that the Q-Score is an effective tool for identifying patients with subclinical rejection who do not show high levels of serum creatinine but have been detected through a protocol biopsy.

The study by Banas et al. [46] investigated the potential of urine metabolites as a biomarker for the noninvasive detection of acute rejection in renal allografts. The researchers analyzed urine samples from 109 renal-transplant recipients and developed a metabolite rejection score. They concluded that the metabolite constellation could serve as a useful biomarker for the noninvasive detection of acute allograft rejection, based on an examination of 46 instances and 520 control samples. Similarly, Chen et al. [47] conducted a study to determine whether urinary CXCL13 levels could be used as a reflection of ongoing immune processes in renal allografts. The researchers quantified urinary CXCL13 levels in 146 renal-allograft recipients and 40 healthy individuals, and found that the method could distinguish between acute cellular rejection and acute antibody-mediated rejection (ABMR) with an area under the curve (AUC) of 0.856. These findings suggest that urinary CXCL13 could serve as a valuable diagnostic indicator for acute rejection in renal allografts. Both Banas et al. [46] and Chen et al. [47] used different methods to analyze urinary metabolites and CXCL13 levels for distinguishing acute rejection from stable groups, affected by various potential confounding factors, such as differences in immunosuppressive treatments, the timing of urine-sample collection, and varying diagnostic criteria for rejection. Patient-specific factors, such as underlying health conditions and genetic variations, can also influence biomarker levels and their predictive accuracy.

In the study conducted by Xu et al. [48], the objective was to identify biomarkers that could effectively predict early acute renal-allograft rejection. The most effective biomarker was found to be a combination of fractalkine on day 0, and IP-10 and IFN-*γ* on day 7. This combination had an area under the curve (AUC) of 0.866, a sensitivity of 86.8%, and a specificity of 89.8%. Zheng et al. [49] utilized gas chromatography–mass spectrometry to distinguish between acute rejection and stable groups by analyzing urine metabolites in 15 individuals who had received renal allografts and were experiencing acute rejection, and 15 individuals who had stable renal transplants. Overall, their study found fourteen metabolites to be significantly different between the acute-rejection group and the stable-transplant group.

Seibert et al. [50] evaluated the diagnostic ability of urinary calprotectin in distinguishing between prerenal and intrinsic acute renal-allograft failure in 328 individuals. The study found that levels of urinary calprotectin were 36 times higher in intrinsic AKI compared to prerenal AKI. Additionally, the ROC curve analysis showed that urinary calprotectin had a high accuracy (AUC = 0.94) in differentiating intrinsic from prerenal AKI. These results demonstrate that urinary calprotectin is able in distinguishing prerenal from intrinsic acute renal-allograft failure. Low levels suggest prerenal AKI and may require immediate fluid repletion, while high levels indicate intrinsic renal failure, possibly necessitating biopsy and specific treatment. However, limitations due to various kidney diseases and infections require further investigation for accurate interpretation. Nonetheless, low-calprotectin concentrations suggest less severe intrarenal damage, while high concentrations warrant further evaluation to rule out other factors.

Viglietti et al. [51] conducted a prospective study involving 851 kidney-transplant recipients to investigate whether systematic monitoring of donor-specific antibodies (DSAs) improves the prediction of kidney-allograft loss. They found that monitoring DSA characteristics and incorporating allograft biopsies into the usual predictors of allograft loss resulted in better accuracy of allograft-loss prediction. Standardized monitoring of anti-HLA DSAs within 2 years post-transplantation, including the IgG3 subclass and complement-binding capacity, enhances allograft-loss prediction beyond conventional approaches. Early detection of subclinical ABMR allows timely intervention and personalized clinical management.

Galichon et al. [52] investigated the potential of urinary mRNA as diagnostic markers for renal-allograft rejection. They analyzed 108 urine samples collected during allograft biopsy and evaluated IP-10 and CD3∈mRNA as potential markers. Normalizing the data based on total RNA quantity did not significantly improve the results, and some conventional reference genes that were overexpressed during rejection even worsened the normalization process. The study highlights the complexities of using urinary mRNA as diagnostic markers for renal-allograft rejection and emphasizes the importance of a urothelial-cell-specific reference gene for accurate normalization. Although the diagnostic value of IP-10 and CD3∈mRNA was not enhanced by reference-gene normalization, the researchers suggest that *GAPDH* and *UPK1A* are preferable reference genes over 18S or HPRT RNA due to their more stable expression levels. These findings offer valuable insights for optimizing normalization methods and improving the reliability of urinary mRNA markers in clinical practice. Venner et al. [53] employed microarray analysis to identify alterations in kidney-transplant biopsies of 315 individuals with pure antibody-mediated rejection (ABMR). They discovered 2603 significantly different transcripts in ABMR biopsies compared to all other biopsies within the total set of 703 biopsies. The study highlighted the importance of transcripts expressed in cultured cell endothelial cells closely related to ABMR. The findings offer valuable insights into using microarray analysis to understand ABMR and provide potential targets for therapeutic intervention, emphasizing the significance of individualized assessment and the need for well-phenotyped biopsy cohorts in modeling human disease states for mechanistic insights.

Shabir et al. [54] studied 73 de novo transplant recipients to explore the link between transitional B lymphocytes and kidney-allograft-rejection rates. They found no significant correlation between transitional B-cells and the development of de novo donor-specific or nondonor-specific antibodies. However, maintaining appropriate transitional B-cell levels was associated with lower rejection rates in patients with de novo donor-specific antibodies. HLA-DR mismatch predicted the time to dnDSA appearance, and prior dnDSA development and repeat transplantation were predictors of dnNDSA appearance. ABMR showed a strong association with microvascular inflammation, and adherent patients with dnDSAs displayed reduced rejection risk. Peripheral cellular regulation may play a role in reducing rejection risk in patients with circulating HLA antibodies. Sigdel et al. [55] utilized iTRAQ-based proteomic discovery and targeted ELISA validation to identify urine protein biomarkers for renal-allograft damage. They found 69 urine proteins with significant differences in abundance between acute rejection (AR) and stable graft, with 12 proteins upregulated in AR and 9 highly specific to AR. These noninvasive biomarkers offer noninvasive advantages over invasive diagnostic methods such as kidney biopsies, aiding risk stratification and personalized management for kidney-transplant patients. The study revealed distinct protein profiles for different injury types (AR, CAI, and BKV), providing insights into underlying mechanisms. Proteins such as PEDF and CD44 show potential as robust AR biomarkers, while common injury-associated proteins suggest autoimmune inflammatory mechanisms in all transplant injuries.

Freitas et al. [56] examined the impact of immunoglobulin-G subclasses and C1q on kidney-transplantation outcomes in recipients with de novo HLA-DQ donor-specific antibodies (DSAs). The study included 284 kidney-transplant recipients who had either persistent DQ-only DSAs or DQ plus other DSAs. Results indicated that individuals with these DSAs had higher rates of acute rejection episodes, allograft loss, and lower 5-year allograft survival rates compared to those without DSAs. Additionally, C1q-positive DSAs were linked to worse pathology and significantly increased graft-loss risk, especially in patients who transitioned from C1q-negative before transplantation to C1q-positive after transplantation. This information aids clinicians in risk stratification and personalized management for transplant recipients. Banasik et al. [57] studied 78 kidney-transplant recipients, with 44% developing de novo donor-specific anti-HLA antibodies (DSAs) within the first year after transplantation. Among DSA-positive patients, seven experienced antibody-mediated rejection, while none of the DSA-negative patients did. The results support prior studies, indicating that newly developed DSAs have a substantial negative effect on graft function and long-term survival. However, caution is necessary due to limitations and potential confounders. While de novo DSAs had a significant impact on graft failure, some patients with DSAs did not experience graft insufficiency during the study, suggesting variability in DSA production and the role of inflammatory events in DSA formation. The mechanism of accommodation and the long-term effects of DSAs on graft outcomes remain unclear, warranting further research. Routine DSA monitoring and early intervention may improve renal-transplant outcomes, requiring better understanding of DSA dynamics beyond the first-year post-transplantation.

Loupy et al. [58] conducted research to investigate the impact of the complement-binding capacity in anti-HLA antibodies on kidney-allograft failure. Their findings suggest that lower graft survival rates associated with complement-binding donor-specific anti-HLA antibodies may be due to complement cascade activation, leading to graft injury and loss independently of C4d deposition, indicating a potential role for complement-dependent pathways in allograft damage. These results highlight the importance of early detection and potential therapeutic interventions targeting the complement to improve graft outcomes in kidney-transplant recipients, which can be translated into clinical practice by using complement-binding donor-specific anti-HLA antibodies as a risk-stratification tool to identify high-risk patients and implement early interventions.

Song et al. [59] analyzed 69 biopsy samples of renal allografts to investigate the clinical relevance of *KIM-1* as a biomarker for tissue damage. They found that *KIM-1* expression was highly positive in the chronic active antibody-mediated rejection group, but weakly positive in the normal group without acute rejection or immunosuppressant toxicity. The results suggest that *KIM-1* expression in renal-allograft-biopsy samples is closely related to markers of tissue damage and rejection, and it may serve as an early marker of rejection injury and graft survival in renal transplantation. Furthermore, *KIM-1* shows potential as an early diagnostic biomarker for acute kidney injury (AKI) and chronic kidney disease (CKD), offering valuable insights for timely intervention and management in clinical settings.

Roshdy et al. [60] conducted a study to determine the association between CRP levels and the early identification of renal-allograft rejection. They monitored 91 renal-transplant recipients for a median follow-up of 8 weeks and discovered that individuals who experienced allograft rejection had significantly higher CRP levels before and after transplantation than those who did not. Hence, the association between CRP levels and early identification of renal-allograft rejection seems significant. Some studies propose CRP estimation as a simple and effective method for detecting rejection and predicting rejection-prone patients. However, other studies suggest the need for considering confounding factors and further research to determine CRP’s reliability as a marker for rejection.

DeVos et al. [61] studied the impact of HLA-DQ donor-specific antibodies on renal-transplantation outcomes. They followed 347 patients for three years and found that 62 patients developed new donor-specific antibodies, with 48 of them having HLA-DQ antibodies either by themselves or in conjunction with other HLA antibodies.

In conclusion, some studies [43,44] reveal promising novel biomarkers for detecting AMR and acute rejection, respectively. Nevertheless, further validation in larger and diverse populations, along with functional experiments, is crucial before clinical implementation. The identified *lncRNAs* demonstrate strong diagnostic accuracy and predictive value for acute rejection compared to traditional clinical markers, but additional validation and comprehensive clinical parameters are necessary for clinical translation. The urinary Q-Score offers advantages over traditional markers, enabling early detection and personalized immunosuppressive therapy. However, challenges in timing and potential influences on metabolite analysis need to be addressed. The biomarker combination of fractalkine, IP-10, and IFN-*γ* shows promise for predicting early acute renal-allograft rejection, but requires multicenter validation and investigation into different rejection scenarios for broader clinical utility (Table 2).

Various studies have been conducted to identify and evaluate different biomarkers for diagnosing acute and antibody-mediated rejection in renal allografts [47,55,57,62]. The biomarkers that were evaluated include proteins, metabolites, long noncoding RNAs, and mRNA. Heidari et al. [43] and Zhang et al. [44] used protein markers, while Banas et al. [46] and Zheng et al. [49] evaluated metabolite markers. Nolan et al. [45] used QSant biomarkers, while Chen et al. [47] examined urinary CXCL13 levels. Xu et al. [50] and Galichon et al. [52] examined a combination of markers, including fractalkine, IP-10, IFN-*γ*, and IP-10 mRNA. Viglietti et al. [51] monitored donor-specific antibodies (DSAs), and Venner et al. [53] examined the expression of transcripts. Lastly, Shabir et al. [54] examined a combination of protein and gene markers. These studies have found promising clinical and combined markers for identifying and predicting acute and antibody-mediated rejection in renal allografts [43,47,53], which could lead to improved diagnosis and treatment outcomes for patients.

## 4. Genetic Markers

Genetic markers have become increasingly important in identifying and managing renal rejection. They provide valuable information about the genetic factors that influence the likelihood of rejection and the response to treatment. Potential confounding factors influencing the association between genetic markers and renal rejection include different gene-expression-analysis technologies (microarray vs. RNAseq), obtaining extra biopsy cores for profiling, and variability in blood tacrolimus concentrations affecting transplant outcomes and AR risk [62]. One of the most critical genetic markers for renal rejection is the HLA mismatch, which refers to differences in human leukocyte antigen between the donor and recipient [63,64]. The HLA system plays a crucial role in presenting antigens to the immune system and is involved in the recognition and rejection of foreign tissues. In addition to the HLA mismatch, single nucleotide polymorphisms (SNPs) in certain genes related to the immune response have been implicated in renal rejection [64]. The presence of specific SNPs in genes involved in immune response can increase the probability of rejection and affect the response to therapy. Gene-expression patterns are another type of genetic marker that has been investigated as a potential indicator of renal rejection [65]. These markers can provide valuable information about the molecular changes that occur during rejection and could be used to develop more targeted treatments. Overall, genetic markers offer significant insights into the mechanisms of renal rejection and provide opportunities for developing personalized treatments [63,64,65].

Wisniewskaa et al. [66] investigated the association between variations in the *VAV1* gene and renal-allograft function. The study found that possessing more T alleles of the *VAV1 rs2546133* variant may have a protective effect against acute rejection in kidney-transplant recipients. This conclusion was drawn from a multivariate regression analysis of 270 patients.

Sommerer et al. [67] aimed to investigate whether monitoring the expression of genes controlled by *NFAT* could serve as an indicator for identifying individuals who are susceptible to developing acute rejection or infections following renal-allograft transplantation. In the study, blood samples from 64 newly transplanted renal-allograft recipients were collected at various intervals after transplantation and subjected to analysis. The findings suggest that individuals with a high level of remaining gene expression are more vulnerable to experiencing acute rejection, whereas those with a low level of remaining gene expression are more susceptible to viral complications, such as the replication of cytomegalovirus and BK virus. Klager et al. [68] examined the diagnostic potential of DARC immunohistochemistry for ABMR by analyzing 82 biopsies from patients who tested positive for donor-specific antibodies (DSAs) and had gene-expression data that could be analyzed. The study revealed a significant association between DARC positivity and ABMR diagnosis, as well as a correlation between DARC gene-expression levels and DARC positivity. However, there was no noticeable difference in graft survival between ABMR cases that were DARC-positive and those that were DARC-negative.

A study conducted by Han et al. [69] investigated the possible connection between dmtDNA levels and antibody-mediated rejection (ABMR) in 323 kidney-transplant recipients. The study found that patients who experienced acute rejection had significantly higher levels of dmtDNA compared to the control group. Multivariate logistic regression analysis demonstrated a significant link between dmtDNA levels and acute rejection. Therefore, measuring the levels of dmtDNA could be a valuable method for predicting acute rejection in individuals who have undergone kidney transplantation. In a study by Groeneweg et al. [70], the influence of acute rejection (AR) on local vascular integrity was examined in 47 kidney-transplant recipients by analyzing kidney biopsies. The researchers observed a reduction in capillary density during AR. Elevated concentrations of LNC-EPHA6 were also identified in the bloodstream during AR, which returned to normal after one year. Furthermore, there was a significant association between the quantities of LNC-RPS24, LNC-EPHA6, and LIPCAR and the marker of vascular damage, soluble thrombomodulin.

Shaw et al. [71] aimed to develop a gene signature capable of tracking acute rejection in kidney transplantation by comparing gene-expression profiles in kidney biopsy and peripheral blood samples from patients experiencing acute rejection (AR) with those of stable patients. They identified a signature of 90 probes, focusing on 76 genes, that could differentiate between stable patients and those experiencing AR. Moreover, they identified a group of eight genes with a unique link to AR. Kim et al. [72] investigated the association between single nucleotide polymorphisms (SNPs) in the genes *EGF* or *EGFR* and the occurrence of end-stage renal disease (ESRD) and acute renal-allograft rejection (AR). Their study included 347 individuals who had received kidney transplants, consisting of 63 patients with AR and 289 healthy individuals. They found that certain SNPs in the genes *EGF* and EGFR were significantly associated with increased susceptibility to both ESRD and AR.

Sharbafi et al. [73] investigated mRNA expressions of *TLR-4*, *TLR-2*, and *MyD88* in PBMCs and biopsy samples of renal-transplant recipients with various types of rejection. The results revealed elevated TLR4 expression in both chronic and acute T-cell-mediated rejection, whereas TLR2 expression was only increased in acute T-cell-mediated rejection. *MyD88* expression was elevated in all types of rejection and could be differentiated from stable grafts. In another study, Guberina et al. [74] examined the correlation between *HLA-E* expression and the survival of renal allografts in the occurrence of acute cellular rejection (ACR). The research outcomes indicated that *HLA-E* expression increased in biopsies with ACR and had a positive correlation with mismatches in HLA-class I leader peptides and the presence of infiltrating cells such as CD8+ and CD56+. The study found that high *HLA-E* expression was linked to reduced allograft survival.

Ge et al. [75] investigated the diagnostic potential of long noncoding RNAs (*lncRNAs*) by analyzing their expression patterns in peripheral blood samples from 150 renal-transplant recipients. The study identified 23 *lncRNAs* with differential expression in both adult and pediatric cohorts, which were able to distinguish between recipients with acute rejection (AR) and those without AR. These findings indicate that these *lncRNAs* can serve as effective diagnostic markers for AR in renal-transplant patients. Qiu et al. [76] examined the role of *lncRNA-ATB* in acute kidney injury (AKI) in renal-transplant patients receiving immunosuppressive therapy. Their study found that *lncRNA-ATB* levels were significantly higher in patients with acute rejection (*n* = 72) compared to the control group. Additionally, their research suggested that *lncRNA-ATB* may affect the characteristics of kidney cells and the toxicity of immunosuppressive medications.

In their study, Liu et al. [77] used next-generation sequencing to detect miRNA expression variations in 15 kidney allografts with acute rejection and normal allografts. Among the 75 dysregulated miRNAs, *miR-10b* had the most significant downregulation in rejected allografts. Inducing a decrease in *miR-10b* expression in human renal glomerular endothelial cells resulted in characteristics similar to those observed in acute allograft rejection. Pawlik et al. [78,79] conducted two separate studies to investigate the association between genetic variations and renal-allograft performance. Using TaqMan genotyping assays and enrolling 270 Caucasian kidney-allograft recipients, the first study did not find any correlation between the Glu37Asp polymorphism in the *renalase* gene and allograft functioning. In contrast, the second study showed that the IVS3 + 17T/C polymorphism in the CD28 gene could have an association with acute rejection, although it did not correlate with delayed graft function or chronic allograft nephropathy.

Kim et al. [80] investigated the relationship between *TLR9* gene polymorphisms and the success of kidney allografts in 342 renal-transplant patients using direct sequencing. They discovered that some alleles of the *TLR9* gene were protective against acute rejection, and two *TLR9* SNPs were linked to the risk of acute rejection in renal transplantation. In a separate study, Suthanthiran et al. [81] developed a noninvasive technique using urine cells to identify acute rejection in kidney transplants. They collected urine samples from 485 kidney-graft patients and analyzed mRNA levels in urinary cells to identify a three-gene signature that could differentiate biopsy specimens with acute cellular rejection from those without it. The estimated AUC from cross-validation was 0.83, indicating good diagnostic accuracy for the three-gene signature in distinguishing biopsy specimens with acute cellular rejection from those without.

The genetic markers related to renal rejection have been validated in some large-scale studies with significant sample sizes, enhancing their robustness and reliability [68,69,72]. These findings hold the potential to transform clinical practice by enabling personalized treatment strategies and improving graft outcomes in kidney-transplant recipients. However, gene-expression patterns as indicators of renal rejection have limitations, and their reliability and consistency across different patients and populations can vary due to genetic diversity and technological differences [67,73]. While the findings from different studies regarding genetic markers for renal rejection align in some areas [75,76], they also present contradictions, with certain markers consistently associated with renal rejection risk, and others showing variable diagnostic potential. Factors such as patient populations, sample sizes, methodologies, and immune response complexity contribute to these discrepancies. Alongside genetic markers, alternative approaches, such as monitoring *NFAT*-controlled genes, investigating *lncRNAs*, and analyzing urinary mRNA levels, offer valuable insights for renal-rejection management. Implementing genetic markers in routine clinical practice faces challenges, including technological differences and the complexity of immune responses, necessitating further research and validation through larger studies for successful integration into transplant management (Table 3).

Several studies investigated the association between genetic markers and the types of rejection in kidney-transplant recipients [67,68,69]. The markers used in these studies vary, including genes (*VAV1*, *NFAT*, *EGF*, *EGFR*), mitochondrial DNA (dmtDNA), long noncoding RNAs (*lncRNAs*), and messenger RNA (mRNA). Despite the variation in the markers, the studies show some common findings. Firstly, they demonstrate that measuring the levels of genetic markers can predict acute rejection in individuals who have undergone kidney transplantation [66,69,70,73]. Secondly, the studies suggest that elevated levels of certain genetic markers are associated with acute rejection, while low levels are associated with viral complications [67,69,80]. Thirdly, the studies reveal that measuring the expression of certain genetic markers can serve as an indicator for identifying individuals who are susceptible to developing acute rejection or infections following renal-allograft transplantation [74,75,77]. Lastly, the studies by Ge et al. and Qiu et al. [75,76] showed that genetic markers, especially IncRNA, can serve as a noninvasive marker for identifying acute rejection in renal-transplant patients.

## 5. Limitations and Strengths

This survey contributes to identifying potential biomarkers and improving diagnostic techniques for acute rejection in kidney transplantation. Some studies have limitations in sample size and diversity, highlighting the need for further validation in extensive and diverse patient populations. However, the studies utilize innovative approaches, such as urine-based tests and T-cytotoxic memory cells, showing promise for personalized medicine and better patient outcomes. These insights can lead to more effective monitoring and management for acute rejection in kidney-transplant recipients. Overcoming limitations requires longitudinal data, optimal monitoring timing, and consideration of confounding variables such as comorbidities and immunosuppressive therapies. Standardizing sample collection methods enhances generalizability. Despite limitations, these studies offer promising efforts to improve acute-rejection monitoring, laying the groundwork for future research. Future studies should focus on validating and exploring clinical, genetic, and histopathological markers for improved kidney-transplant outcomes. Noninvasive early detection markers, specific biomarkers for different rejection types, and personalized immunosuppressive therapies show potential. Validation, standardization, and longitudinal studies are essential to address knowledge gaps and enhance patient care. Collaboration and ongoing research are vital to overcome immune rejection challenges. Moreover, widespread implementation of these markers in clinical practice may require further validation and replication studies to ensure their reliability and reproducibility across different populations and healthcare settings. Additionally, the adoption of histopathological markers for renal-transplant rejection may also depend on the availability and expertise of pathology resources in various medical centers. To increase the clinical applicability of these markers, ongoing research and collaboration between researchers, clinicians, and pathologists are essential.

## 6. Conclusions

In conclusion, various histopathological, clinical, and genetic markers have been identified as important indicators for identifying different types of renal rejection in kidney-transplant recipients. T-cell-mediated rejection (TCMR) and antibody-mediated rejection (ABMR) were found to be common in acute rejection, and detecting and treating acute rejection was found to be critical in improving treatment outcomes. Biomarkers, including proteins, metabolites, long noncoding RNAs, and mRNA, have been evaluated to diagnose acute and antibody-mediated rejection in renal allografts, showing promising results in improving diagnosis and treatment outcomes. Genetic markers, including genes, mitochondrial DNA, long noncoding RNAs, and messenger RNA, were also found to be important in predicting acute rejection in kidney-transplant patients and identifying individuals susceptible to developing acute rejection or infections following renal-allograft transplantation. Measuring the expression of certain genetic markers, especially IncRNA, has been identified as a noninvasive marker for identifying acute rejection in renal-transplant patients. The evaluation of these markers provides valuable insights into the mechanisms underlying kidney-transplant rejection and may lead to improved treatment outcomes for kidney-transplant patients.

## Figures and Tables

**Figure 1 biomedicines-11-02437-f001:**
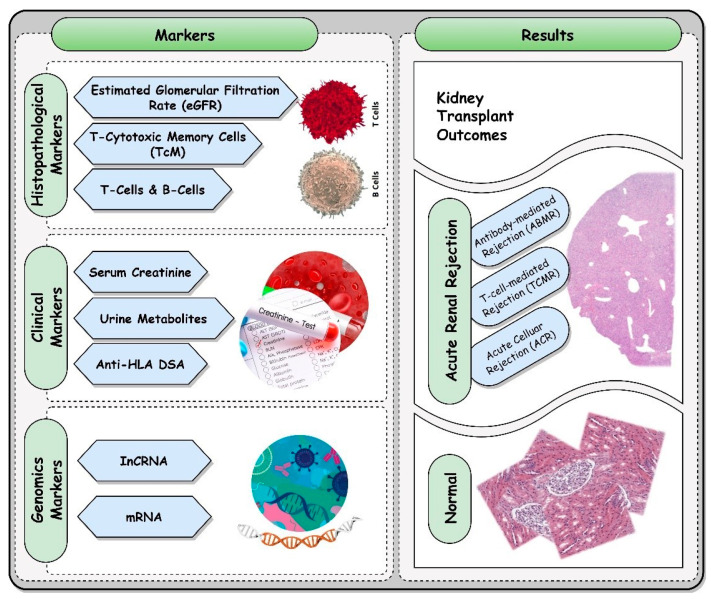
A brief overview of different markers utilized in the early identification of acute renal rejection.

**Table 1 biomedicines-11-02437-t001:** A review of the existing literature regarding histopathological markers used to anticipate acute rejection in patients.

Reference	Aims	Features	Approach	Results	Outcomes
Arai et al. [13]	Evaluate the clinicopathological results of **269 cases** of acute/active antibody-mediated rejection (AABMR).	**Clinicopathological:** 6 markers **Histological features:** peritubular capillaritis, glomerulitis, and interstitial inflammation **Immunological features:** donor-specific antibodies (DSAs) and C4d staining	**Histological:** light microscopy **Immunological:** ultrastructural examination	29 patients had PAABMR diagnoses Average g-score value of 1.17 ± 0.6 Mean ptc score was 1.97 ± 1.32 69% of PAABMR were DSA-positive	The protocol biopsy mostly revealed PAABMR.
Garcia-Covarrubias et al. [14]	Examine the urine renal cells of **37 patients** with the diagnosis of humoral rejection to determine the mRNA-profile-expression pattern for interleukin (IL-8) and transform growth factor β1.	**Histopathological slides:** mRNA from biopsies and urine samples	Cross-sectional analytical study	The renal rejection group’s urine samples indicated an overexpression of the proinflammatory IL-8. 10 patients had a diagnosis of humoral rejection	IL-8 mRNA may serve as a diagnostic tool when there is a persistent damage caused by fibrosis.
Eiamsitrakoon, H. et al. [15]	Obtain precise estimate about the influence of interferon gamma D874 T > A (*rs2430561*) polymorphism on renal-allograft rejection.	**Histopathological slides:** (*IFNG*) *β*1 874T/A(*rs2430561*)	Meta-analysis—eighteen articles (twenty-two studies) were reviewed	An allele polymorphism was associated with a heightened susceptibility to renal rejection (odds ratio = 1.43, *p* = 0.0004, 95 percent confidence interval: 1.18–1.74).	Interferon gamma + 874 T > A polymorphism can serve as a useful indicator for identifying patients who are at an elevated risk of experiencing renal-allograft rejection.
Rohan et al. [16]	Assess the efficacy of allospecific TcM at regular clinical visits for **22 renal-transplant patients**.	**Histopathological slides:** T-cytotoxic memory cells (TcMs)	Serial samples were collected between 3 and 4 weeks Blood samples were drawn in sodium heparin tubes	T-cell-mediated rejection (TCMR) was confirmed in six out of twenty-two patients. Antibody-mediated rejection (ABMR) was shown in the seventh patient. Out of fifteen patients, ten tested negative, which translates to a specificity of 67% (specificity 67%). 5 patients had false-positive tests	Individuals with a dysfunctional transplant were distinguished from primary kidney-transplant patients with stable transplants by the presence of allospecific T-cytotoxic memory cells.
Shimizua et al. [17]	Examine the borderline changes (BCs) that occurred after **22 renal transplantations** and explore the potential clinical or pathological significance of these changes.	**Clinicopathological:** analyses of borderline changes (BCs)	The clinical and pathological data of cases exhibiting borderline changes were analyzed retrospectively	No renal grafts were lost throughout the observation period, but nine patients experienced a decline in renal-allograft function following the biopsy.	Given that nearly half of the patients experienced a decline in renal graft function, BCs might have played a role in acute T-cell-mediated rejection.
Zhuang et al. [18]	Analyze the distinctions in B-cell subsets between **73 immunologically stable kidney-transplant cases** and **103 cases** with end-stage renal disease (ESRD).	**Histopathological slides:** B-cell subsets	Flow cytometry	Kidney-transplant patients exhibited lower percentages of B-cells (CD19 + f) and regulatory B-cells (Breg) overall. Renal-allograft recipients had lower percentages of transitional B-cells and marginal zone (MZ) B-cells.	Patients with ESRD had minimal B-cell subsets, different from those immunologically stable kidney-allograft recipients.
Mueller et al. [19]	Use RNA sequencing on **34 kidney-transplant-biopsy specimens**; 18 were normal subjects and Banff acute T-cell-mediated rejection (TCMR) was diagnosed in 16 cases.	**Combined markers:** 39 genes **Innate immune and system transcriptome marker** cytokines, chemokines, *Toll-like receptors*, *interferon-response genes*, and complement components T-cell activation and inflammation	RNA sequencing Computational analysis of intragraft mRNA transcriptome	TCMR biopsies had a higher intragraft levels of calcineurin mRNA, suggesting under-immunosuppression compared with normal biopsies.	High innate immune system gene expression was identified using transcriptome profiling during a TCMR episode in kidney allografts.
Nowanska et al. [20]	Examine ETAR immunoreactivity significance in **162 patients** with renal-allograft biopsies because of transplant-function degradation.	**Combined markers:** endothelin A receptors (ETARs) associated with either acute tubular necrosis or antibody-mediated rejection	Microscopic evaluation	Patients who tested positive for ETARs had a higher incidence of graft loss within one year after biopsy, but statistical significance was not reached. There was no statistical significance achieved; however, graft loss was identified in 1-year patients who tested positive for ETARs after biopsy.	Endothelin receptor expression in renal blood vessels appears to be significant in determining the extent of injury caused by acute tubular necrosis and antibody-mediated rejection.
De Leur et al. [21]	Hypothesize that acute T-cell-mediated rejection in **15 patients** can be regulated by T-cells, such as interleukin (IL)-211B cell lymphoma 6 (BCL6)1 T follicular helper cells, within the allograft.	**Histopathological slides:** T- and B-cells	Immunohistochemical staining. Statistical analysis: descriptive statistics, chi-squared test, analysis of variance. Histological examination: hematoxylin and eosin staining, multiple xIHC, and in situ hybridization.	ELSs with an active phenotype were primarily discovered in aTCMRI regions where T-cells and B-cells co-localize.	Organized lymphoid structures in acute T-cell-mediated rejection indicate that T- and B-cell interactions may have a significant role in this type of renal-allograft rejection.
Wu et al. [22]	Verify the hypothesis that a human-kidney-biopsy sample may include a complete description of the cell types and states using single-cell RNA sequencing.	**Combined markers:** endothelial cells, dendritic-cell maturation	Single-cell transcriptomics RNA sequencing	Genes related to activation of the Fc receptor pathway and the internalization of Ig were expressed by one set of activated endothelial cells, supporting the pathologic diagnosis of antibody-mediated rejection.	New segment-specific proinflammatory responses in rejection were discovered when the transcriptomes of healthy kidney epithelial tissue and their equivalents from biopsy specimens were compared.
MCRAE et al. [23]	Find out whether **17 patients** with end-stage renal failure who undergo transplantation and experience acute cellular rejection demonstrate expression of CD4+ CD25+/CD39+ in their peripheral blood.	**Combined markers:** T-cell markers—CD4, CD25, CD39	Flow cytometry Statistical analyses	On biopsy, patients with acute TCMR showed lower levels of CD4+ CD25+/CD39+ mTreg and CD4+ CD25+/CD39+ mTeff cells. Long-term transplant recipients had lower levels of CD4+ CD25+/CD39+ mTreg and CD4+ CD25+/CD39+ mTeff compared to nonimmunosuppressed controls.	Reduced mTreg and mTeff levels were present in acute cellular rejection.
Jiang et al. [24]	Analyze the effect of CD20-positive B-cell infiltration in biopsy specimens obtained from 216 individuals who experienced acute cellular rejection (ACR) in their transplanted organ.	**Pathological:** CD20-positive B-cell	Histological analysis: biopsies were immunostained for CD20 and C4d Statistical analysis: Cox regression analysis	Patients who had CD20-positive B-cell invasion in their allograft biopsy samples had lower estimated survival rates at the 1-year, 3-year, and 5-year marks compared to those who did not have CD20-positive infiltration.	CD20-positive B-cell infiltration during acute cellular rejection (ACR) may be indicative of unfavorable outcomes for allografts.
Sentis et al. [25]	Investigate the predictive value of glomerular leukocyte infiltration during episodes of acute renal-allograft rejection in a sample of **57 patients**.	**Clinicopathological:** 6 markers **Histological features:** estimated glomerular filtration rate (eGFR), glomerulitis, interstitial inflammation, peritubular capillaritis, and interstitial inflammation **Clinical markers:** serum creatinine levels and proteinuria	Immunohistochemistry: kidney biopsies Statistical analyses: Cox and logistic regression models	During acute renal rejection, there was an association of a high number of glomerular infiltrating leukocytes (GILs) with a worse prognosis for graft survival.	GILs, to be specific, T-cells, could be beneficial as a prognostic marker for acute renal rejection.
Matignon et al. [26]	Perform a study to analyze the T-cell immune response using biopsies from **43 patients** with acute T-cell-mediated rejection.	**Combined markers:** TH-17 alloimmune responses **Clinicopathological markers:** donor age, donor type, ischemia time, delayed graft function, and estimated glomerular filtration rate	Immunohistochemistry Gene-expression analysis	Patients who showed the presence of mRNA IL17 and low-mRNA Foxp3 expressions had higher rates of treatment failure compared to patients without high-mRNA expression of IL17 or high-mRNA expression of Foxp3.	Individuals who have received expanded-criteria allografts demonstrate involvement of the Th17 pathway in the development and outcome of acute T-cell-mediated rejection.
Visona et al. [27]	Understand the function of lymphomononuclear inflammation (nephritis) in the renal-allograft medulla among a group of **113 patients** experiencing acute dysfunction.	**Pathological:** anti-CD4, CD8, CD20, CD68, and CD138 antibodies, B lymphocytes, macrophages, and plasmocytes, and cytotoxic T-cells, and T-helper cells	Immunohistochemistry (IHC)	Out of the total 113 cases observed, 66 cases showed corticomedullary nephritis, while the remaining 47 cases had exclusively medullary nephritis.	The immunophenotype of medullary nephritis in renal-allograft biopsies may serve as a potential indication of acute cellular rejection (ACR).
Lee et al. [28]	Examine the effect of directed antibodies against AT1R (anti-AT1R) in **53 patients**.	**Clinicopathological markers:** serum angiotensin II type 1 receptor antibodies **Histopathological features:** donor-specific HLA antibodies (DSAs) and anti-AT1Rs	Histopathological examination ELISA assay Statistical analysis: multivariate logistic regression, correlation analysis, Mann–Whitney U test	DSAs and HLA antibodies were identified in 75.5%, and 49.1%, respectively. The rejection group had a 9.4% prevalence of anti-AT1R.	The presence of anti-AT1R and DSAs was associated with patients who had antibody-mediated rejection (ABMR) in their renal allografts.
Crespoet al. [29]	Investigate the association between nondonor-specific and donor-specific anti-HLA antibodies with peripheral blood NK-cell subsets and clinical characteristics in a group of **393 patients**.	**Combined markers:** serum creatinine levels, anti-HLA donor-specific antibodies (DSAs) **Histopathological features:** NK-cell antibody-dependent cell-mediated cytotoxicity (ADCC)	Enzyme-linked immunosorbent assay (ELISA) Flow cytometry Statistical analysis: logistic regression analysis, Mann–Whitney U-test	The percentage of CD56(dim) NK cells was found to be higher in individuals who experienced acute rejection as compared to those who did not, according to the analysis. Patients with acute rejection had a lower proportion of CD56(bright) NK cells compared to those without rejection.	Analyzing the immune characteristics of natural killer (NK) cells may help to identify the alloreactive humoral response patterns in recipients of kidney transplants.
Bhat et al. [30]	Evaluate the clinical significance of functionally active p-S6RP plasma cells in ARin **28 renal allografts**.	**Histopathological features:** p-S6RP plasma cells	Immunohistochemistry staining Statistical analysis: *t*-test, Mann–Whitney test, Kaplan–Meier analysis, log-rank test	Following treatment, the high-score group had a higher p-S6RP staining and CrR (*p* < 0.05) than the low-score one. There was no notable distinction in the CrR (clinical response rate) between the two groups when it came to the staining of CD20 or CD138.	Plasma cells that secrete p-S6RP antibodies with functional activity are often linked to an inadequate treatment response and commonly involved in allergic rhinitis.
DeVos et al. [31]	Detect the influence of acute rejection and DSAs on intermediate-term graft loss in **227 patients**.	**Combined markers:** histological features, association of acute rejection, and donor-specific antibody	Retrospective analysis Statistical analysis: log-rank test, Kaplan–Meier survival curves, logistic regression	Out of the total individuals observed, 24% developed a de novo donor-specific antibody (dDSA), and of these individuals, 73% had dDSA targeting the DQ antigen.	The emergence of dDSAs was linked to a higher occurrence of graft loss, but the negative impact of dDSAs was restricted to individuals experiencing acute rejection in the intermediate term.
Zhao et al. [32]	Examine the role of soluble fibrinogen-like protein 2 (*sFGL2*) in the acute rejection of human kidney transplants and how it is controlled in a group of **40 patients**.	**Clinicopathological markers:** serum levels of *soluble FGL2*	Flow cytometry ELISA Western blotting Real-time PCR Cell-culture techniques	Individuals who experienced acute rejection (AR) following a kidney transplant exhibited elevated levels of sFGL2, TNF-a, IFN-g, and CD4+ T-cells in their bloodstream.	There is a possibility that *sFGL2* could serve as a biomarker for the acute rejection of transplanted organs and may also contribute to the progression of this condition.
Ge et al. [33]	Investigate the association between the *IFNG +874 T >* A gene variation and acute rejection (AR) following a kidney transplant in a cohort consisting of **525 patients** with AR and **1126 patients** without AR.	**Histopathological features:** interferon gamma (*IFNG) +874 T > A*	Meta-analysis study	The *IFN-gamma +874 T > A* gene variation was significantly associated with a higher likelihood of acute rejection of transplanted kidneys, particularly among individuals of Asian descent.	Individuals who underwent kidney transplantation and received a cadaveric kidney transplant were found to have a higher risk of acute rejection if they carried the *IFNG +874 T > A* gene variation.
Loupy et al. [34]	Investigate whether the ability of anti-HLA antibodies to bind to complements is associated with kidney-transplant failure in a sample of **1016 patients**.	**Combined markers:** complement-binding anti-HLA antibodies and kidney-allograft survival, histological features	Retrospective analysis Statistical analysis: Cox proportional-hazards regression, Kaplan–Meier survival, multivariable analysis, logistic regression	Individuals who had donor-specific anti-HLA antibodies that can bind to complements had the lowest 5-year graft-survival rate compared to those who had donor-specific anti-HLA antibodies that cannot bind to complements or those without donor-specific anti-HLA antibodies.	Assessing the ability of donor-specific anti-HLA antibodies to bind to complements appears to be useful in identifying patients who are at a greater risk of experiencing kidney-transplant failure.
Li et al. [35]	Identify the inflammatory cells involved in BK virus nephropathy (BKVN) and evaluate the efficacy of immunophenotyping in differentiating between BK virus nephropathy (BKVN) and acute rejection (AR) in **65 patients**.	**Histopathological features:** immune cells, including CD4, CD8, CD20, CD68, and FOXP3	Immunostaining Immunophenotyping	There were notable disparities in the number and percentages of CD3, CD4, CD8, and CD20 cells between BKVN and AR, while there were no discrepancies in the expression of HLA-DR in tubule cells.	Renal biopsies’ CD20 cell count and CD3, CD4, CD8, and CD20 cell percentages could help differentiate BKVN and AR. The increase in HLA-DR expression may not only signify acute rejection but could also be a reaction to BK virus nephropathy (BKVN).
Xu et al. [36]	Analyze the immune phenotype of T-lymphocyte infiltrations in surveillance of renal biopsies with stable renal function early post-transplantation of **242 patients**.	**Clinicopathological:** regulatory/cytotoxic infiltrating T-cells	Immunohistochemistry Statistical analysis: univariate and multivariate logistic regression, and Cox regression	The cytotoxic phenotype group includes 16.8% of cases, all of which developed biopsy-proven acute rejection or clinical-diagnostic acute rejection within 1 year after biopsy.	Analyzing the immunophenotype of T-cells that infiltrate the transplanted kidney in early post-transplant biopsies could predict the occurrence of acute rejection and the patient’s long-term survival.
Chang et al. [37]	Examine a total of **56 renal-transplant biopsies** which were classified based on the Banff schema into three categories: T- cell-mediated acute rejection (21 cases), antibody-mediated acute rejection (18 cases), and mixed acute rejection (17 cases).	**Clinicopathological markers:** plasma cell densities, glomerular filtration rates (GFR)	Biopsy analysis: immunohistochemistry Statistical analysis: univariate Cox proportional hazard analysis	In all types of acute rejection, CD3+ T-cells were found to be the predominant cell type, followed by CD20+ B-cells and CD138+ plasma cells. The density of plasma cells was a notable predictor of graft failure, whereas the density of B-cells exhibited a tendency towards significance.	Plasma cells could play a crucial role as a mediator or a sensitive marker of steroid-resistant acute rejection.

**Table 2 biomedicines-11-02437-t002:** A review of the existing literature regarding clinical markers used to anticipate acute rejection in patients.

Reference	Aims	Features	Approach	Results	Outcomes
Heidari et al. [43]	Examine the urine proteome of **36 patients** to identify new diagnostic biomarkers for AMR.	**Urinary:** urinary epidermal growth factor (*EGF*)	Proteome analysis Random forest (RF) algorithm	Sensitivity 77% Specificity 68% (AUC) 0.984	The analysis of urinary proteomics in patients with AMR suggests that urinary *EGF* could potentially serve as an early diagnostic biomarker for AMR.
Zhang et al. [44]	Identify new long noncoding RNAs (*lncRNAs*) that can be used as diagnostic markers for acute rejection (AR) to predict the risk of graft loss in a cohort of **282 patients**.	**Clinical:** long noncoding RNAs (*lncRNAs*)	Quantitative real-time PCR (qRT-PCR) Statistical analyses: logistic regression, univariate and multivariate Cox regression	Sensitivity 0.836 Specificity 0.733 AUC = 0:891	Discovered four new long noncoding RNAs (*lncRNAs*) that could potentially serve as biomarkers for identifying acute rejection (AR) in renal allografts.
Nolan et al. [45]	Develop and validate a urinary Q-Score for determining confirmed acute rejection biopsy in **223 patients**.	**Combined markers:** 5 markers **Urinary:** chemokines, LIM domain only protein 7 (LMO7), and periostin **Clinical:** creatinine and proteinuria	Quantitative test QSant Transcriptomic analysis of urine samples	The Q-Score, combining molecular features, was effective in identifying and distinguishing antibody-mediated rejection, T-cell-mediated rejection, and stable nonrejecting patients.	QSant is a precise and quantitative method appropriate for regular surveillance of the status of a kidney transplant.
Banas et al. [46]	Confirm the effectiveness of a new method for monitoring kidney-transplant rejection that does not require invasive procedures in a cohort of **109 patients**.	**Combined markers:** 6 urinary metabolite constellations	Mass spectrometry Liquid chromatography techniques Statistical analyses: Kruskal–Wallis tests, Bonferroni correction, multivariate logistic and linear regressions, and Mann–Whitney U test	Found that combining the metabolite rejection score with the estimated glomerular filtration rate (eGFR) at the time of urine sampling significantly improved the overall test performance. This was evidenced by an area under the curve (AUC) of 0.84. The study involved 109 patients.	The use of NMR-based urine metabolite analysis and eGFR together has potential to be a useful method for post-transplant surveillance and to assist in histopathological evaluation.
Chen et al. [47]	The objective of the study was to investigate if urinary C-X-C motif chemokine 13 (CXCL13) could serve as an indicator of immune processes in **146 renal-allograft patients and 40 healthy individuals**.	**Clinicopathological markers:** urinary C-X-C motif chemokine 13 (CXCL13)	Immunohistochemical analysis Statistical analyses: Kaplan–Meier survival analysis, logistic regression, and Cox proportional hazards model	Observed that urinary CXCL13/creatinine levels were lower in normal transplants than in cases of acute tubular necrosis, chronic allograft nephropathy, and acute rejection. As a result, the area under the curve (AUC) for acute rejection was 0.818.	Increased urinary levels of CXCL13/Cr were associated with a decreased response to steroid treatment and impaired function of the transplanted kidney.
Xu et al. [48]	Explore the possibility of utilizing a group of serum biomarkers as a combined tool to predict acute rejection in renal allografts in a sample of **99 patients**.	**Combined markers:** 3 markers Serum biomarkers (fractalkine, IFN-γ and IP-10)	Enzyme-linked immunosorbent assay (ELISA) Statistical analysis: Mann–Whitney U test, and logistic regression analysis	The best prediction of acute renal-allograft rejection was obtained by combining fractalkine on the first day, IP-10 on the seventh day, and IFN-γ on the seventh day, with an AUC of 0.866. Sensitivity 86.8% Specificity 89.8%.	Using a combination of the identified biomarkers has the potential to enhance the early detection of acute renal-allograft rejection.
Zheng et al. [49]	Create noninvasive techniques to precisely detect acute rejection in **15 patients** who received renal allografts and are experiencing rejection, and **15 stable patients**.	**Combined markers:** urine metabolites	Gas chromatography–mass spectrometry (GC/MS) recipients Statistical analysis: leave-one-out analysis and partial least squares (PLS)	Identified metabolites included glycerol, glycolic acid, citric acid, malic acid, fructose, and proline s had a high accuracy for detecting acute rejection with an AUC = 0.914.	The identification of specific metabolites in urine samples can help differentiate between individuals with acute rejection and those with stable transplants, suggesting that analyzing the metabolome could be a useful noninvasive method for diagnosing acute rejection.
Seibert et al. [50]	Examine if calprotectin could differentiate between prerenal and intrinsic acute kidney injury (AKI) in a group of **328 patients**.	**Clinicopathological markers:** Urinary calprotectin.	Enzyme-linked immunosorbent assay (ELISA) Statistical analysis: receiver operating characteristic (ROC) curve analysis	Analysis of the ROC curve indicated that calprotectin (AUC = 0.94) has a high level of accuracy in distinguishing between intrinsic and prerenal AKI. Sensitivity 90.4% Specificity 74.1%	Urinary calprotectin is an increasingly used biomarker for distinguishing between prerenal and intrinsic acute renal-allograft failure.
Viglietti et al. [51]	Investigate if the predictive accuracy for kidney-allograft loss among a group of **851 patients** can be improved by systematic monitoring of donor-specific antibodies.	**Clinical:** donor-specific anti-HLA antibodies (anti-HLA DSAs).	Luminex single-antigen bead assay and complement-dependent cytotoxicity assay Statistical analysis: univariate and multivariate Cox regression	Of the patients who received a transplant, 110 (12.9%) were found to have DSAs, and after transplantation, screening revealed that 186 (21.9%) patients were DSA-positive.	The monitoring and characterization of DSAs before and after transplantation may enhance individual risk stratification for kidney-allograft loss.
Galichon, et al. [52]	Investigate urinary mRNA to diagnose renal-allograft rejection involved analyzing 108 urine samples collected during allograft biopsy, including **56 patients** without acute rejection and **52 with acute rejection**.	**Combined markers** 9 markers **Clinical:** estimated glomerular filtration rate, serum creatinine levels, and proteinuria **Urinary:** urine mRNA levels of six genes	RT-quantitative PCR (qPCR) Statistical analysis: Wilcoxon and Kruskal–Wallis tests, chi-squared test, logistic regression models	Additional normalization did not result in improvement in the overall RNA quantity and may even exacerbate the issue, causing overexpression of classical reference genes during rejection.	The use of a reference gene for normalization is crucial in maintaining the quality and reproducibility of polymerase chain reaction (PCR) and mitigating the impact of RNA degradation.
Venner et al. [53]	Determine the modifications in pure ABMR among **315 patients** in the discovery set and **264 patients** in the validation set.	**Clinical:** endothelial cells, CD16a Fc receptors	Gene-expression profiling Flow cytometry Statistical analysis: *t*-tests, ANOVA, and linear regression analysis	The transcript changes in ABMR that were observed in the initial discovery set were highly consistent in a subsequent validation set.	The alterations in transcripts related to ABMR that were detected in the original discovery set exhibited a high degree of similarity in a subsequent validation set.
Shabir et al. [54]	Determine the potential association between transitional B lymphocytes and protection against kidney-allograft rejection in a cohort of **73 patients**.	**Clinical:** transitional B lymphocytes	Flow cytometry In vitro assays	The presence of transitional B-cells (as opposed to total B-cells or regulatory T-cells) was associated with a decreased risk of acute rejection.	Transitional B-cells have the potential to serve as a biomarker in transplantation due to their capacity to reveal the interrelationships between humoral immunity, cellular immunity, and nonadherence.
Sigdel et al. [55]	Identify possible biomarkers for urine protein from a group of **262 patients** with biopsy-confirmed allograft injury.	**Combined markers:** 3 markers **Clinical:** donor age and glomerular filtration rate (eGFR) **Proteomic:** urine proteins	iTRAQ—proteomic discovery Targeted ELISA and immunohistochemistry	Among the urine proteins studied, 69 exhibited differences in levels between acute rejection cases and other transplant categories. Of these, 9 urine proteins were significantly specific to acute rejection due to variations in other transplant categories.	The fibrinogen proteins effectively differentiated between acute renal-allograft rejection and BK virus nephritis, providing support to use noninvasive diagnostic methods.
Freitas et al. [56]	Identify characteristics of complement-binding for the most harmful DQDSA in **284 kidney-transplant recipients**.	**Clinicopathological markers:** 11 markers **Clinical:** human leukocyte antigen mismatches, age, gender, ethnicity, biopsy, panel-reactive antibodies, serum creatinine, proteinuria, DSAs **Pathological:** C1q in de novo HLA-DQ donor-specific antibody and immunoglobulin-G subclasses	Enzyme-linked immunosorbent assay (ELISA) Flow cytometry Histological analysis Statistical analysis	Individuals with de novo DQ-only and acute rejection demonstrated elevated levels of antibodies that combine IgG1/IgG3 and bind to C1q. The presence of de novo DQ donor-specific antibodies that bind to C1q was associated with a 30% decrease in allograft survival at the 5-year mark.	Persistent de novo DQ donor-specific antibodies that bind to the complement have a negative impact on kidney-allograft outcomes.
Banasik et al. [57]	Analyze the long-term survival and graft function in a group of **78 patients** who developed de novo donor-specific anti-HLA antibodies following transplantation.	**Clinical:** de novo donor-specific anti-HLA antibodies	Complement-dependent lymphocytotoxic technique Immunological assays	The existence of post-transplant donor-specific antibodies had a significant impact on both graft survival and kidney function, although, in 38% of cases, it did not result in a reduction of renal function after five years.	Having stable renal-allograft function in the presence of donor-specific antibodies after transplantation is a positive indication of long-term allograft survival and function.
Loupy et al. [58]	Examine if anti-HLA antibody complement-binding capacity is important for renal-allograft failure for **1016 patients**.	**Clinicopathological markers:** kidney-allograft survival and complement-binding anti-HLA antibodies	Solid-phase assays and flow cytometry Banff classification Statistical analysis: Cox proportional-hazards regression	The presence of donor-specific anti-HLA antibodies that bind to the complement after transplantation was associated with a high incidence of antibody-mediated rejection.	Assessing the ability of donor-specific anti-HLA antibodies to bind to thew complement is advantageous in identifying individuals who are at risk for renal-allograft loss.
Song et al. [59]	Examine the expression of kidney injury molecule-1 (*KIM-1*) and evaluate its clinical relevance as a biomarker for tissue damage in a group of **69 patients**.	**Clinical:** kidney injury molecule-1 (KIM-1)	Immunohistochemistry Light microscopy Statistical analysis	There was a positive correlation between *KIM-1* expression and the severity of tubular inflammation in the acute T-cell rejection category. There was a positive expression of *KIM-1* in the group with chronic active antibody-mediated rejection.	*KIM-1* expression is a potential and early biomarker for identifying injury to renal tubular epithelial cells in samples from kidney allografts.
Roshdy et al. [60]	Examine the correlation between CRP and the early detection of acute renal-allograft rejection in a group of **91 kidney-transplant recipients** as a predictive and emerging marker.	**Clinicopathological markers:** CRP (C-reactive protein) and acute renal rejection	Retrospective analysis of medical records Triple-drug-immunosuppressive treatment Statistical analysis: logistic regression, chi-square tests, and *t*-tests	In individuals with allograft rejection, both before and after transplantation, CRP levels were elevated compared to those who did not experience rejection.	A high serum CRP level prior to transplantation may be a useful predictor for monitoring post-transplant patients.
DeVos et al. [61]	Examine the impact of donor-specific HLA-DQ antibodies on the outcomes of renal transplantation in a group of **347 recipients**.	**Combined markers:** donor-specific HLA-DQ antibodies, age, gender, race, and underlying kidney disease	Serological analysis of pre- and post-transplant serum samples Statistical analysis: Kaplan–Meier survival analysis, and univariate and multivariate Cox proportional hazards regression	When the combination of HLA-DQ antibodies with non-DQ antibodies, alone or the absence of antibodies (92–94)	The detection of donor-specific HLA-DQ antibodies was the most common, and it was associated with poorer graft outcomes.

**Table 3 biomedicines-11-02437-t003:** A review of the existing literature regarding genomic markers used to anticipate acute rejection in patients.

Reference	Aims	Features	Approach	Results	Outcomes
Wisniewskaa, I. et al. [66]	Investigate the relation between *VAV1* gene polymorphisms and kidney-allograft function in **270 patients**.	**Genetics:** *VAV1* gene polymorphisms	Genotyping Statistical analysis: logistic regression models, and chi-square test	A greater number of *VAV1 rs2546133* T alleles exhibited a protection against acute rejection among recipients of kidney transplants. No correlations were observed between *VAV1* genotypes and chronic allograft dysfunction or delayed graft function.	Variations in the *VAV1* gene have been linked to the rejection of kidney allografts.
Sommerer, et al. [67]	Evaluate *NFAT*-regulated gene expression in renal-allograft recipients to detect those of acute rejection in **64 patients**.	**Combined markers:** gene expression of several genes, including CYP3A5, CYP3A4, IL2RB, and CD3E	Gene expression microarray analysis Statistical analysis: correlation analysis, *t*-tests, ANOVA, and linear regression analysis	Patients with high residual gene expression had increased risk of acute rejection, while those with low residual gene expression had a higher viral complications incidence.	*NFAT-RGE* was confirmed to be a potential noninvasive biomarker to identify acute-rejection patients.
Klager et al. [68]	Assess the ability of DARC to serve as a diagnostic marker for ABMR in a cohort of **741 patients**.	**Genetics:** Duffy antigen receptor for chemokines (DARC)	Immunohistochemical staining Pathological analysis Statistical analysis: linear regression models, Wilcoxon signed-rank test, and Mann–Whitney U test	Detection of DARC was linked to ABMRdiagnosis and demonstrated correlation with expression levels of the DARC gene, determined by microarray analysis.	DARCscore is a beneficial method to diagnose and monitor subclinical ABMR in kidney-transplant patients.
Han et al. [69]	Investigate the association between the levels of dmtDNA and antibody-mediated rejection (ABMR) occurrence in **323 patients**.	**Combined markers:** 7 markers **Genetics:** donor plasma mitochondrial DNA (mtDNA) **Clinicopathological:** estimated glomerular filtration rate, urine protein-to-creatinine ratio, pretransplant donor-specific antibodies (DSAs), donor age, human leukocyte antigen mismatches, and serum creatinine	Real-time quantitative PCR Single-antigen bead assay (SAB) Statistical analysis: logistic regression, and Kaplan–Meier survival analyses	Renal-transplant recipients who developed antibody-mediated rejection (ABMR) were found to have higher levels of donor plasma mitochondrial DNA (mtDNA) than those who did not develop ABMR.	The possibility exists that the levels of donor mtDNA may function as a biomarker for forecasting the likelihood of AMR in recipients of renal transplants.
Groeneweg et al. [70]	Examine the impact of acute rejection (AR) after renal transplantation on local vascular integrity and evaluate the levels of circulating four *lncRNAs* **47 patients**.	**Combined markers:** 9 markers **Genetics:** LNC-EPHA6, a circulating *long noncoding RNA* **Clinical:** cause of end-stage renal disease, donor and recipient age, ethnicity, gender, type of donor, type of immunosuppressive therapy, HLA mismatches, and time after transplantation	RNA isolation and qRT-PCR Statistical analysis: logistic regression analysis	Patients who developed T-cell-mediated AR following renal transplantation were found to have increased levels of LNC-EPHA6 in their circulation.	The increased levels of LNC-EPHA6 in patients who experienced AR, as opposed to those who did not, provide support for the idea that *lncRNAs* could be used as indicators or markers for AR.
Shaw et al. [71]	The objective was to devise a gene signature that can diagnose AR across different immunosuppressive regimens, independent of age, in a cohort of **110 patients**.	**Combined markers:** age-independent gene signature	RNA extraction and gene-expression profiling Statistical analysis: Mann–Whitney U Tests	A subset of eight genes (DIP2C, ENOSF1, FBXO21, KCTD6, PDXDC1, REXO2, HLA-E, and RAB31) was identified as being linked to AR.	A novel gene network was discovered that was independent of age and could identify AR in both blood and kidney samples.
Kim et al. [72]	Examine whether there is a possible correlation between single nucleotide polymorphisms (SNPs) in the genes encoding epidermal growth factor (EGF) or its receptor and either end-stage renal disease (ESRD) or acute allograft rejection (AR) in a cohort of **347 patients**.	**Combined markers:** *EGF* receptor gene polymorphisms and epidermal growth factor (EGF)	Genotyping Statistical analysis: logistic regression	The assay using SNPs in dd- cfDNA was capable of distinguishing between active rejection and nonrejection with an AUC of 0.87, 88.7% sensitivity, and 72.6% specificity.	There may be an association between SNPs in the *EGF* and *EGFR* genes and the development of ESRD and AR.
Sharbafi et al. [73]	Assess the levels of TLR-4, *TLR-2*, and *MyD88* mRNA expressions in biopsy samples and peripheral blood mononuclear cells (PBMCs) in a group of **50 patients** with different types of rejection.	**Combined markers:** myeloid differentiation primary response 88 (*MyD88*) genes, Toll-like receptor 4 (*TLR-4*), and Toll-like receptor 2 (*TLR-2*)	RNA isolation and cDNA synthesis Real-time PCR Statistical analysis: Kruskal–Wallis test	Both ACMR and CCMR cases showed elevated levels of TLR4 mRNA. The upregulation of TLR2 gene expression was only in patients with ACMR. MyD88 expression was able to differentiate stable grafts from AAAMR, ACMR, CAMR, and CCMR cases.	Assessment of the expression level of inflammatory signaling genes may have predictive value to identify the type of allograft rejection.
Guberina, et al. [74]	Examine the impact of *HLA-E* on the survival of acute cellular rejection in a group of **25 individuals**.	**Combined markers:** *HLA-E* expression during acute cellular rejection, renal-allograft survival, HLA class I leader peptide mismatches	Immunohistochemistry HLA typing Statistical analysis: Kaplan–Meier curves and Cox proportional hazards regression analysis	During acute cellular rejection, high expression of *HLA-E* and presence of HLA class I leader peptide mismatches were linked to lower renal-allograft survival.	*HLA-E* could be used to improve risk stratification and personalized treatment for kidney-transplant patients.
Ge, et al. [75]	Examine the patterns of *long noncoding RNA (lncRNA)* expression in the peripheral blood (PB) of **150 renal-transplant recipients**.	**Combined markers:** two (*lncRNAs*) *long noncoding RNAs: NONHSAT076754 and TCONS-00007075*	Quantitative real-time PCR (qRT-PCR) Statistical analysis: univariate and multivariate logistic regression analyses, and Cox regression	The study created a risk score based on the expression levels of the two most significant *lncRNAs (AF264622 and AB209021)* and found that it had excellent diagnostic accuracy for both recipients of primary renal transplantation (PRTx) and those who underwent allograft retransplantation (ARTx).	The two lncRNA molecular signatures in PB might function as an innovative noninvasive biomarker for AR diagnosis.
Qiu et al. [76]	Examine the long noncoding RNA function named ATB in acute rejection and its effect on postoperative pharmaceutical immunosuppression therapy in a sample of **108 patients**.	**Genetics:** (*lncRNA-ATB*) long noncoding RNA	Real-time quantitative PCR Enzyme-linked immunosorbent assay (ELISA) Immunohistochemistry Statistical analysis: logistic regression, Kaplan–Meier method	*LncRNA-ATB* was found to be significantly upregulated in patients with acute rejection.	Significant changes in the expression of *lncRNA-ATB* were observed in patients with acute rejection, suggesting its potential as a new biomarker for the detection of this condition.
Liu, et al. [77]	Investigate the role of microRNA-10b (*miR-10b*) in **15 acute rejections of renal allografts** and its relationship with the expression of the pro-apoptotic gene BCL2L11.	**Genetics:** microRNA-10b and BCL2L11	Reverse transcription polymerase chain reaction (RT-PCR) Immunohistochemistry Flow cytometry Statistical analysis: ANOVA	MicroRNA-10b targeted and suppressed the expression of BCL2L11, a proapoptotic gene, and that decreased microRNA-10b levels led to increased BCL2L11 expression and apoptosis in renal allografts.	Targeting microRNA-10b and BCL2L11 could serve as a strong therapeutic strategy to prevent acute rejection in patients.
Pawlik et al. [78]	Examine the relationship between kidney-allograft function and the Glu37Asp polymorphism in the *renalase* gene (rs2296545) in a cohort of **270 patients**.	**Genetics:** *renalase* gene polymorphism	DNA isolation and genotyping Statistical analyses: logistic regression and chi-squared test	The rs2296545 polymorphism in the *renalase* gene was not found to be significantly associated with delayed graft function or acute rejection in the studied cohort.	The rs2296545 polymorphism in *the renalase* gene does not appear to be a determinant of renal-allograft function.
Pawlik, et al. [79]	Assess the relationship between CD28 gene IVS3 + 17T/C (rs3116496:T/C) polymorphism and delayed renal-graft function (DGF), and acute-rejection development, in **270 patients**.	**Genetics:** CD28 gene	DNA isolation and genotyping Statistical analysis: logistic regression analysis	No statistically significant associations were observed between CD28 gene polymorphism and delayed functioning of a transplanted kidney or chronic allograft nephropathy.	The IVS3 + 17T/C CD28 gene polymorphism has the potential to be used as an indicator to predict acute rejection in people who have undergone kidney transplantation.
Kim et al. [80]	Explore the correlation between *TLR9* polymorphisms and renal-allograft outcomes in **342 patients**.	**Clinicopathological markers:** *TLR9* gene polymorphisms (*rs187084 and rs352140*), estimated glomerular filtration rate (eGFR)	Genotyping Statistical analyses: chi-square test, logistic regression analysis	The presence of *rs187084* TT genotype was linked to a higher susceptibility to acute rejection, while having *rs352140* AA genotype was linked to a lower susceptibility to acute rejection.	*TLR9* gene polymorphisms have the potential to serve as genetic markers for predicting acute rejection and renal function in recipients of renal transplants.
Suthanthiran et al. [81]	Evaluate the effectiveness of a noninvasive diagnostic test that uses mRNA profiles in urinary cells to detect acute cellular rejection at an early stage in a group of **485 individuals** who received kidney transplants.	**Combined markers:** 18 different genes **Urinary-cell mRNA markers:** CD3*ε* mRNA, IP-10 mRNA, 18S rRNA **Clinicopathological markers:** immunosuppressive therapy, serum creatinine, histologic diagnosis	RNA extraction and microarray analysis Gene-set enrichment analysis Statistical analysis: linear regression, univariate analyses, Cox proportional-hazards regression, multivariate logistic regression models, and Kaplan–Meier survival analysis	The urinary-cell mRNA profile outperformed the current clinical method for detecting ACR, with a sensitivity of 80% and a specificity of 94%. The ability of the mRNA profile to differentiate ACR from other causes of graft dysfunction was shown to have a sensitivity of 75% and specificity of 95%.	A noninvasive gene-expression assay that measures the expression of three mRNA transcripts (18S rRNA, IP-10, and CD3*ε* mRNA) can accurately identify patients with acute cellular rejection (ACR) of kidney allografts.

## Data Availability

Not applicable.

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
