# Peer review of "Biomarkers for Kidney-Transplant Rejection: A Short Review Study"

_biomedicines, 2023, doi:10.3390/biomedicines11092437_

Round 1
Reviewer 1 Report
Introduction
Can you provide more specific information about the sources and reliability of the survival rate data mentioned in reference [1]?
In reference to immune rejection, what are the main factors contributing to graft rejection, apart from genetic dissimilarity? Are there any advancements or potential solutions in this area?
The involvement of T lymphocytes, innate and adaptive immune systems, cytokines, and costimulatory molecules in rejection is mentioned. Could you elaborate on the relative significance of each of these components and their interactions?
How common is hyper-acute rejection compared to other types of acute renal rejection? Is it possible to provide some statistical data or references to support its severity and occurrence rate?
In terms of the mentioned markers (histological, clinical, and genetic), which ones have shown the most promise in detecting acute renal rejection in its early stages? Are there any comparative studies or data available to support their effectiveness?
The introduction mentions the correlation between biomarkers and the type of acute renal rejection. Can you provide more details on how these markers differentiate between antibody-mediated rejection (ABMR) and T-cell-mediated rejection (TCMR)?
When discussing the structure of the review, could you clarify if there are any specific criteria or selection process used to choose the studies included in each section (histopathological, clinical, genetic markers)?
Extraction of RNA and Clinical Markers from Histological Samples
How do the histopathological markers discussed in the text compare to clinical markers in terms of accuracy and reliability? Are there any studies that directly compare the two types of markers?
Are there any conflicting or contradictory findings among the studies mentioned in the text? If so, what are the implications of these discrepancies?
Are there any studies or research mentioned in the text that have not been replicated or validated by other researchers? How does the lack of replication affect the reliability of the findings?
What are the sample sizes of the studies mentioned in the text? Are the sample sizes large enough to draw meaningful conclusions?
How applicable are the findings discussed in the text to real-world clinical practice? Have these markers been widely adopted and implemented in the diagnosis and monitoring of renal transplant rejection?
Clinical Markers
How reliable are the identified clinical markers (serum creatinine levels, BUN levels, urine output, proteinuria) in detecting renal rejection? Are there any limitations or potential sources of error associated with these markers?
What are the specific advantages and limitations of using biopsy or imaging in conjunction with clinical markers to confirm a diagnosis of rejection?
Regarding the novel biomarkers identified by Heidari et al. and Zhang et al., what are the sample sizes of their studies, and have these biomarkers been validated in larger cohorts or different populations?
How do the identified long noncoding RNAs (lncRNAs) and urinary Q-Score compare to traditional clinical markers in terms of diagnostic accuracy and predictive value for acute rejection?
What are the potential challenges or limitations in implementing the use of urinary metabolites as biomarkers for non-invasive detection of acute rejection?
Regarding the study by Xu et al., how applicable and generalizable are the findings of the identified biomarker combination (fractalkine, IP-10, IFN-γ) to different patient populations or transplant centers?
What are the potential confounding factors or variables that might influence the analysis and interpretation of urinary metabolites in distinguishing between acute rejection and stable groups?
How reliable is urinary calprotectin as a diagnostic marker for differentiating between prerenal and intrinsic acute renal allograft failure? Are there any limitations or potential sources of error associated with its use?
In the study by Viglietti et al., how did the incorporation of monitoring donor-specific antibodies (DSA) improve the prediction of kidney allograft loss compared to conventional predictors? Were there any specific characteristics of the DSA that were found to be particularly informative?
What are the specific challenges or considerations in using urinary mRNA as diagnostic markers for renal allograft rejection? How do the findings of Galichon et al. and Venner et al. contribute to our understanding of this approach?
How well do transitional B lymphocytes correlate with rejection rates in patients who developed de novo donor-specific antibodies? Are there any potential confounding factors or alternative explanations for the observed relationship?
Regarding the proteomic analysis conducted by Sigdel et al., what are the practical implications of the identified urine protein biomarkers for renal allograft damage? How do these biomarkers compare to existing diagnostic methods?
What are the implications of the impact of immunoglobulin-G subclasses and C1q on kidney transplantation outcomes in recipients with de novo HLA-DQ donor-specific antibodies? How can these findings guide clinical management and treatment strategies?
How do the effects of de novo donor-specific anti-HLA antibodies on graft function and survival, as investigated by Banasik et al., align with previous research in this area? Are there any limitations or potential confounders that should be considered?
What are the potential mechanisms or underlying reasons for the lower graft survival rates associated with complement-binding donor-specific anti-HLA antibodies, as demonstrated by Loupy et al.? How can these findings be translated into clinical practice?
How does the expression of KIM-1 in biopsy samples of renal allografts relate to other markers of tissue damage or rejection? What are the potential applications of KIM-1 as a biomarker in clinical settings?
How strong is the association between CRP levels and the early identification of renal allograft rejection? Can CRP levels be used as a reliable and specific marker for rejection, or are there confounding factors that need to be considered?
Genetic markers
What are the potential confounding factors that could influence the association between genetic markers and renal rejection?
Have these genetic markers been validated in large-scale studies or only investigated in small sample sizes?
How do the findings regarding genetic markers for renal rejection translate into clinical practice? Are there any specific interventions or treatments that can be developed based on this research?
Are there any ethical considerations or potential implications associated with using genetic markers for identifying individuals susceptible to rejection or infections following renal transplant?
What are the limitations of using gene expression patterns as indicators of renal rejection? How reliable and consistent are these markers across different patients and populations?
How do the findings from different studies regarding genetic markers for renal rejection align or contradict each other?
Are there any alternative or complementary approaches to identifying and managing renal rejection that should be considered alongside genetic markers?
What are the challenges and feasibility of implementing the measurement of genetic markers in routine clinical practice?
Conclusion
What are the future directions for research in this area? What are the gaps in knowledge or areas that require further investigation to enhance the understanding and application of these markers in kidney transplant recipients?
-
Reviewer 2 Report
This review paper was written without a clinically important concept.
The paper cited plenty of studies, but most of them do not present important data. In a consequence the reader is loosing time.
In general clinicians have 3 important issues: 1. Replacing kidney graft biopsy, 2. Limitations related to scoring of bordelnine Banff cases, 3. Differentiation between acute and chronic rejection, especially in cases with potential overlap.
These aspect should be completely separated from predisposing factors.
The lecture of the texts does to try to answer these 3 questions.
Reviewer 3 Report
This is a nice paper. However, I have some comments. The findings from this paper are excellent and worthy to review. This manuscript contained some questions described below. I think this paper is interesting, this review contributes to future's clinical medicine largely. I have some questions from a point of view of clinical medicine. This paper contains very detailed information on rejection in relation to renal transplantation. Please provide additional information on clinical markers used in practice. From the view of the pathology described by the authors, we consider the use of tubular injury markers to be clinically important. Please add a note on the usefulness of β2-microglobulin, NGAL, NAG, KIM1 and L-FABP as markers of rejection in urine tests used in real clinical practice.
Reviewer 4 Report
This review is comprehensive and the text reads well. I have only minor comments.
Introductions needs to be shortened in particular in its first general part
The clinical markers needs to be named rather biochemical serum and urine markers and may be better divided into biochemical serum and urine, hormonal and combined.
the English language quality is sufficient
